# Unbroken $\mathscr{PT}$-symmetry in the absence of gain or loss

Johannes Bentzien [1], Julien Pinske[2], Lukas J. Maczewsky[1], Steffen Weimann[1], Matthias Heinrich[1], Stefan Scheel[1] & Alexander Szameit[1] ✉

The concept of parity-time symmetry has firmly established non-Hermiticity as a versatile degree of freedom on a variety of physical platforms. In general, the non-Hermitian dynamics of open systems are perceived to be inextricably linked to complex-valued potentials facilitating the local attenuation and coherent amplification in wave mechanics. Along these lines, time reversal symmetry is associated with a complex conjugation of the potential landscape, in essence swapping gain and loss. Here we leverage nonorthogonal coupled-mode theory to synthesize genuinely non-Hermitian dynamics without either gain or loss, and experimentally demonstrate parity-time symmetry via fluorescence measurements in femtosecond-laser-written arrays. Our projective approach allows features of non-Hermiticity to be utilized in scenarios where actual amplification and/or attenuation may disrupt the desired physics, e.g. in nonlinear systems or quantum optics.

The conventional approach to implement non-Hermitian systems is the introduction of its corresponding imaginary potential by deliberately engineering interactions with an environment (Fig. 1a). The Hamiltonians of such systems remain symmetric and generally have complex eigenvalues[1,2], except in scenarios where some underlying symmetry of the system ensures a global cancellation of gain and loss, e.g., for parity-time ($\mathscr{PT}$) symmetric configurations in the unbroken phase[3]. The fascinating physics of $\mathscr{PT}$ systems have been explored on a number of experimental platforms ranging from microwave cavities[4] and photonic waveguides[5] to electric circuits[6], ultra-cold atoms[7] and even coupled mechanical pendulums[8]. Experiments with $\mathscr{PT}$-symmetric quantum systems have been conducted with single photons[9], NV centers in diamond[10], or superconducting qubits[11], and second-quantization effects in a $\mathscr{PT}$ system were recently observed for photon correlations in lossy waveguide couplers[12]. In the context of quantum systems, the presence of actual gain inevitably gives rise to noise, which, in most cases, prevents a direct adaptation of "classical" non-Hermitian settings[13]. While many features of non-Hermitian systems persist when a global loss is imposed[14], the exponential decay of modes that have real eigenvalues in the co-damped frame may limit the practical reach of such passive $\mathscr{PT}$-systems.

Here, we devise an entirely different approach that allows for non-Hermitian dynamics to be observed without actually changing the total intensity of light. Instead, we employ projections in a nonorthogonal basis to leverage systematic deviations from conventional orthogonal coupled-mode theory. This approach intermittently conceals certain parts of an evolving wave function in a way that establishes genuinely non-Hermitian dynamics for the light residing on the vertices of discrete optical systems. The underlying concept is illustrated in Fig. 1b, taking a coupled set of three mechanical oscillators as an example. At rest, the equidistant vertices of this setting lie on a straight line in three-dimensional space (akin to a triatomic linear molecule such as $CO_2$), and dynamical deflections from their respective equilibrium positions along and perpendicular to the symmetry axis are schematically represented by semitransparent ghosts. An orthogonal projection, corresponding to a light source at infinite distance aligned perpendicular to the symmetry axis, faithfully preserves the oscillation amplitudes in the shadows cast onto a two-dimensional screen oriented perpendicular to the light ray direction (left). In contrast, when placing a point source at any finite distance and/or at a different angle, the oscillation amplitudes are, in general, no longer preserved: shadow oscillators' ghosts are neither equidistant nor of the same size (right). In particular, the latter reveals the projection-induced non-Hermiticity

[1]Institute of Physics, University of Rostock, Albert-Einstein-Str. 23, 18059 Rostock, Germany. [2]Niels Bohr Institute, University of Copenhagen, Blegdamsvej 17, DK-2100 Copenhagen, Denmark. ✉e-mail: alexander.szameit@uni-rostock.de

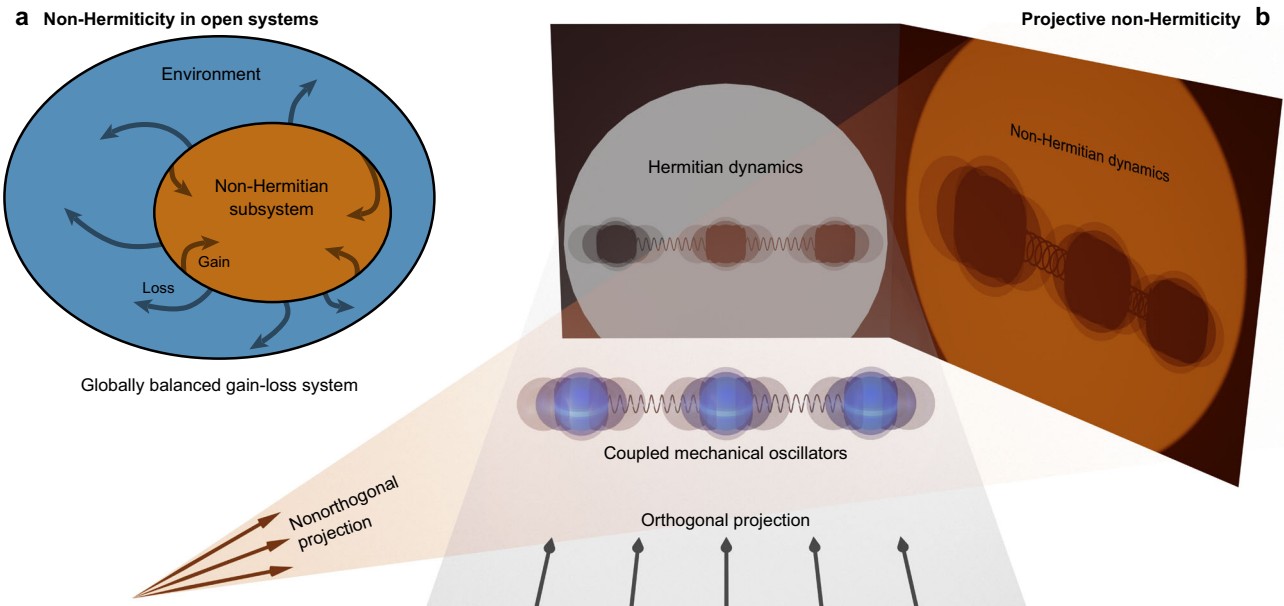

**Fig. 1 | Different approaches to non-Hermiticity. a** Conventional gain-loss systems: In a globally balanced gain-loss system, the overall population is conserved, while the subsystem is subject to a non-Hermitian evolution. **b** Novel design via nonorthogonal projection explicitly avoiding gain and loss: A Hermitian system of three coupled mechanical oscillators is freely oscillating in three-dimensional space, which is schematically represented by semitransparent ghosts. A light source placed at infinity perpendicular to the oscillators' symmetry axis corresponds to an *orthogonal projection*, as the oscillation amplitudes in the shadows cast onto a perpendicular screen are preserved (left). Any other placement of the light source represents a *nonorthogonal projection*, i.e., shadows are in general no longer equidistant or of the same scale (right). In particular, the latter exemplifies a projection-induced non-Hermiticity due to the nonorthogonal basis, as the amplitudes of the oscillation of the spheres are now rendered anisotropically, and the total mechanical energy of the system therefore appears to be no longer conserved.

of this system due to the nonorthogonal basis: the amplitudes of the oscillation of the spheres are now rendered anisotropically, and the total mechanical energy of the system therefore would appear to be no longer conserved, although the conservation of the original system's total mechanical energy obviously still holds true. If the structure itself is $\mathscr{PT}$-symmetric and the nonorthogonal projection is preserving this property, one can even use the stricter term "projective $\mathscr{PT}$-symmetry" rather than "projective non-Hermiticity" to elucidate the distinction from implementations of $\mathscr{PT}$-symmetry involving gain or loss. Note that, in the absence of any actual gain or loss, projective $\mathscr{PT}$-symmetry is impervious to spontaneous breaking. In contrast to complex-valued potentials, Hamiltonians generated by projective $\mathscr{PT}$-symmetry are in principle asymmetric. Furthermore, our approach allows for the generation of $\mathscr{PT}$-symmetric Hamiltonians that extend beyond the trivial scenario where $\mathscr{PT}$-symmetry is achieved in every Hermitian matrix that is real and symmetric.

With this mechanical example of the general concept of $\mathscr{PT}$-symmetry in mind, let us turn to the optical context, where we consider systems of evanescently coupled single-mode waveguides. When fabricated by means of femtosecond laser pulses in fused silica[15], fluorescence imaging[16] in such structures allows us to quantitatively observe the local intensity in the core of each waveguide, thereby implementing the desired nonorthogonal projection onto the lattice sites. In the following, we will show theoretically and demonstrate experimentally that the degree of nonorthogonality can be seamlessly tuned via the inter-waveguide spacing, and readily allows for $\mathscr{PT}$-symmetric wave dynamics to be observed in settings devoid of gain or loss.

## Results

### Nonorthogonal coupled-mode theory

The propagation of coherent light in arrays of $N$ coupled waveguides is governed by the paraxial Helmholtz equation (see "Methods", Eq. (3)) or, by applying the tight-binding ansatz (5) in terms of the amplitudes

$(a_k(z))_{k=1}^{N}$ of the individual transverse waveguide modes $w_k(x,y)$. In this discretized picture, the propagation dynamics is mediated by the interplay of the evanescent couplings $c_{k,l}$ between waveguides $k,l$ and their respective propagation constants $\beta_k$, leading to the discretized coupled-mode equation[17]

$$i P \partial_z \mathbf{a} = -K \mathbf{a} \qquad (1)$$

Here, the real-valued $N \times N$ matrix $K = C + PB$ is comprised of the propagation constants ($B$) and the couplings ($C$), as well as the real-valued matrix $P$ representing the power overlap of the waveguide modes. In the context of conventional orthogonal coupled-mode theory (OCMT), this latter contribution is assumed to be negligible ($P = \mathbb{I}_{N \times N}$). By contrast, whenever the transverse mode profiles of adjacent waveguides do indeed measurably overlap ($P \neq \mathbb{I}_{N \times N}$), non-orthogonal coupled-mode theory (NOCMT) has to be employed to extend the reach of the tight-binding picture (see "Methods", Eq. (5)) to encompass such configurations. To this end, similar to the coupling coefficients, the overlaps are determined from the waveguide modes $w_k(x,y)$. The underlying assumption of this ansatz is that these modes are localized around their corresponding waveguide and therefore unaffected by the presence of the other waveguides, allowing for their calculation as solutions of Eq. (3) (see "Methods") for the refractive index profile of only the $k$-th waveguide[18].

A more flexible approach is to substitute the waveguide mode amplitudes $a_k(z)$ by modal amplitudes $(b_k(z))_{k=1}^{N}$ derived by transverse normal modes $v_k(x,y)$[19]. As generalizations of the waveguide modes $w_k(x,y)$, the normal modes $v_k(x,y)$ can be obtained as suitably localized superpositions of the full system's solutions (supermodes, see Fig. 2a–c) of the paraxial Helmholtz equation according to Eq. (6). As such, the normal modes $v_k(x,y)$ in general can feature non-zero contributions from the other waveguides of the system, allowing them to form a set of orthonormal eigenfunctions even under conditions where the $w_k(x,y)$ no longer do. While the two approaches converge in

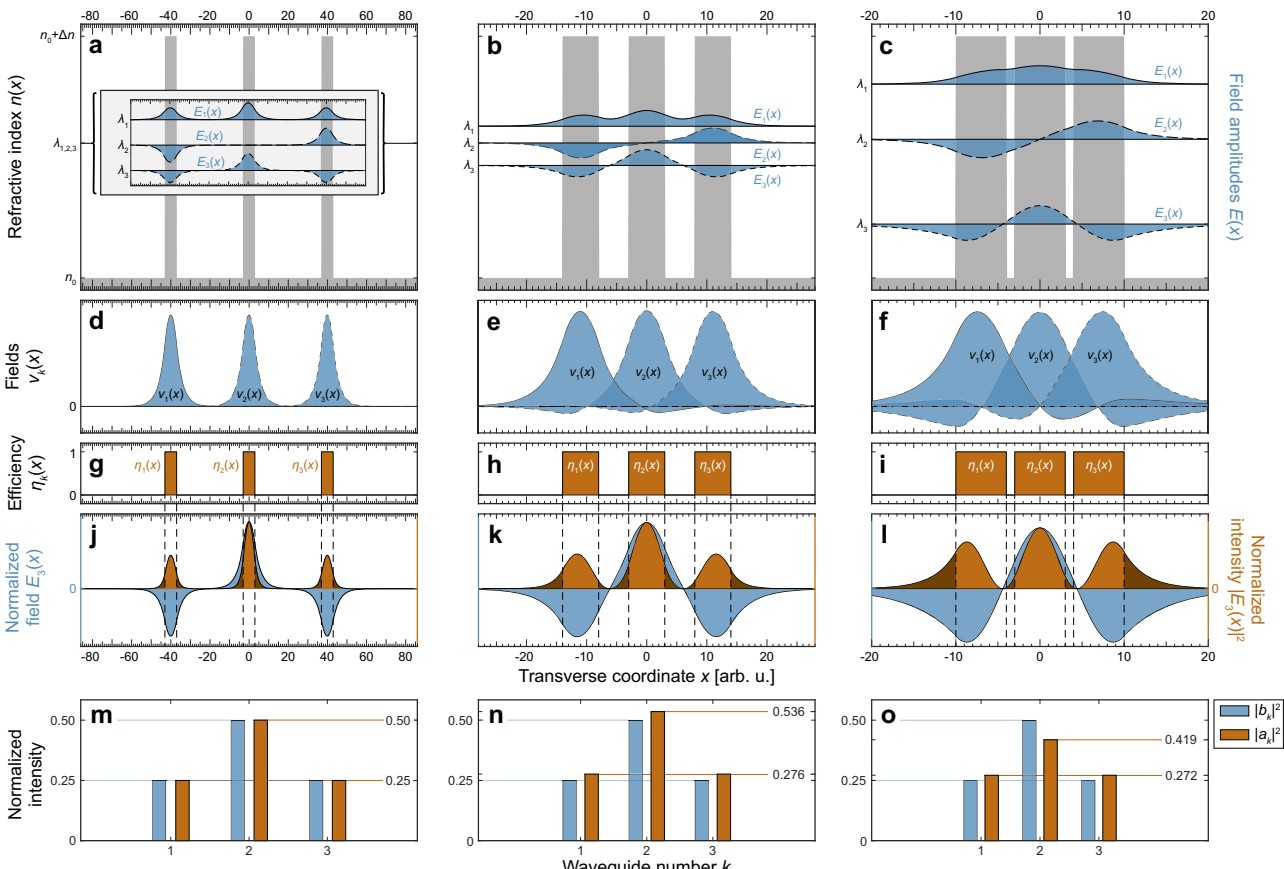

**Fig. 2 | Projective non-Hermiticity in coupled-mode theory and fluorescence measurements. a–c** Refractive index landscape (gray) of three identical step-index waveguides and resulting set of supermodes $E_k(x)$ for different center-to-center waveguide spacings $d = 40$ arb. u. (**a**), $d = 11$ arb. u. (**b**) and $d = 7$ arb. u. (**c**). The vertical position of the supermodes indicates their respective eigenvalues $\lambda_k$ (see Eq. (4), "Methods"). Deviations from zero amplitude in the minima of the first supermode are a measure for the degree of nonorthogonality of the system and become more pronounced as the spacing is reduced. **d–f** Normal modes $v_k(x)$ constructed from superpositions of the supermodes $E_k(x)$ (see "Methods", Eq. (6)). For large waveguide separations (**d**), the normal modes $v_k(x)$ become indistinguishable from the parity-symmetric modes $w_k(x)$ of hypothetical isolated waveguides at the same position. For smaller waveguide separations (**e**, **f**), $v_k(x)$ and $w_k(x)$ diverge as the normal modes incur contributions from the other waveguides as well. While normal modes are by definition always orthogonal ($\int v_k^* v_j \mathrm{d}x\mathrm{d}y = \delta_{jk}$), the nonorthogonality of the system is embodied by the waveguide modes

($\int w_k^* w_j \mathrm{d}x\mathrm{d}y \neq \delta_{jk}$). **g–i** As a simplified model for the waveguide fluorescence mechanism, the fluorescence efficiency $\eta_k(x)$ can be assumed to be identical to the index profile, i.e., uniform inside and zero outside the guides. **j–l** As shown here for the example of $E_3(x)$, fluorescence imaging therefore does not measure the overall intensity distribution (dark orange) but rather the on-site fraction of it (orange). **m–o** In the coupled-mode approximation, propagation dynamics are determined by the amplitudes $a_k$ of the waveguide modes $w_k(x)$. In contrast to the normal mode occupations (blue bars), which in this system are $(|b_k|^2)_k = (\frac{1}{4}, \frac{1}{2}, \frac{1}{4})$ regardless of the waveguide spacing, the on-site intensities $|a_k|^2$ (orange bars) are proportional to the overlap with the fluorescent core regions and therefore may dynamically vary upon propagation as in general $\sum_k |a_k|^2 \neq 1$. In the orthogonal regime (left column), $a_k$ and $b_k$ are identical as $w_k(x)$ and $v_k(x)$ become indistinguishable. In the opposite limit (right column), the conventional coupled-mode approximation eventually breaks down, and the arrangement becomes better represented by a single three-mode waveguide.

the limit of weakly interacting waveguides (see Fig. 2d), the differences between the tight-binding and normal mode expansions are revealed when the inter-waveguide minima in the first supermode deviate from zero (see Fig. 2b, c). Note that in our example of parity-symmetric waveguides, this coincides with a breaking of parity symmetry of the normal modes in line with their different environments as determined by their respective positions within the structure (see Fig. 2e, f). More details of the coupled-mode theory are provided in "Methods"/Supplementary Note 1.

Note that Eq. (1) resembles a discrete Schrödinger equation with a Hamiltonian $H = P^{-1}K$. It follows that, if $P$ and $K$ are Hermitian (i.e., symmetric for real-valued matrices) and do not commute, the real-valued non-symmetric matrix $H$ is necessarily non-Hermitian[20] (for the proof see Supplementary Note 1). Moreover, the normal mode expansion $\mathbf{b} = Q\mathbf{a}$[17,21] maps any Hamiltonian that can be represented by our projective approach onto a Hermitian Hamiltonian $\widetilde{H} = (Q^{-1})^\dagger K Q^{-1}$. Here, the transformation $Q$ is an $N \times N$ matrix that

factorizes the power matrix via $P = Q^\dagger Q$ (see Supplementary Note 2 for more information). The non-Hermitian nature of the Hamiltonian $H$ results in a non-unitary evolution where the on-site intensity $I = \sum_k |a_k|^2$ is no longer a conserved quantity, despite the fact that, in the absence of any gain or loss, the overall intensity $\widetilde{I} = \sum_k |b_k|^2$ remains constant during its evolution governed by the Hermitian Hamiltonian $\widetilde{H}$.

## Connection between theory and fluorescence measurements

Far from being a purely theoretical construct, the on-site intensity is directly accessible via waveguide fluorescence microscopy in laser-written photonic lattices, as this technique utilizes color centers formed in the focal volume (i.e., the waveguide core) during the inscription process[16]. In this vein, the negligible fluorescence of the surrounding host material allows for quantitative observations of discrete propagation dynamics with a high signal-to-noise ratio[15]. The

individual on-site intensities $|a_k|^2$ are proportional to the overlap between the local fluorescence efficiency $\eta_k$ (Fig. 2g–i) and the propagating mode ($\int \eta_k |E(z)|^2 \mathrm{d}x\mathrm{d}y \propto |a_k(z)|^2$, cf. orange areas Fig. 2j–l and orange bars in Fig. 2m–o). Analogously, the overlap between the normal modes and the propagating mode gives rise to the normal mode amplitudes ($|\int v_k^* E(z)\mathrm{d}x\mathrm{d}y|^2 = |b_k(z)|^2$, see Supplementary Note 2). While the latter are not individually accessible in the experiment, the overall intensity $\sum_k |b_k|^2$ can be observed by near-field measurements at the output facet of the sample. To illustrate the difference between normal and waveguide modes, let us consider the third supermode as an example for the propagating mode in Fig. 2j–l. The corresponding normal mode occupations are $(|b_k|^2)_k = (1/4, 1/2, 1/4)$, i.e., the overall intensity $\sum_k |b_k|^2 = 1$ is conserved in all cases (Fig. 2m–o, blue bars). The mode occupations of the other two supermodes are provided in Supplementary Note 2. In conventional settings, i.e., for well-separated waveguides with orthogonal modes, the on-site fluorescence faithfully tracks the overall intensity of the wave function with a constant scaling factor related to the fraction of the overall intensity occupying the space at the lattice sites.

This manifests itself in equal total fluorescence efficiencies of the three supermodes at large waveguide separations (blue part of Fig. 3b). In contrast, under nonorthogonal mode conditions, the total fluorescence efficiency dynamically changes during propagation in line with the evolving interference pattern between the supermodes (orange part of Fig. 3b). In other words, fluorescence imaging maps the set of normal modes $(v_k(x,y))_k$ onto the set of waveguide modes $(w_k(x,y))_k$ in a nonorthogonal fashion. For even closer waveguides,

the conventional coupled-mode approximation eventually breaks down, the second supermode's total fluorescence efficiency increases (white part of Fig. 3b), and the former arrangement of three single-mode waveguides becomes better represented by a single three-mode waveguide.

## Projective $\mathcal{PT}$-symmetry

Importantly, non-Hermitian Hamiltonians based on modal nonorthogonality can readily emulate $\mathcal{PT}$-symmetric systems despite not being symmetric (i.e., $H \neq H^{\mathrm{T}}$). As the time-reversal operator $\mathcal{T}$ (i.e., complex conjugation, $\mathcal{T}^2 = \mathbb{I}_{N \times N}$) acts trivially on the real-valued entries of $H$, the condition for $\mathcal{PT}$-symmetry simplifies to $H = \mathcal{P}H\mathcal{P}$. Moreover, in place of an actual parity flip, any suitable permutation matrix fulfilling $\mathcal{P}^2 = \mathbb{I}_{N \times N}$ can be chosen such that $[H, \mathcal{PT}] = 0$ is satisfied throughout the evolution[22]. Note that projective non-Hermitian systems converge to the conventional Hermitian case as the waveguide modes become orthogonal by increasing the separation between the waveguides, or, in terms of the above formalism, if $P$ and $K$ commute. Supplementary Fig. 5 illustrates how the Hamiltonians embodied by NOCMT and discrete $\mathcal{PT}$-symmetric systems in general are embedded within the different types of complex $N \times N$-matrices. This Venn diagram also completes Fig. 1 by Bender et al.[22].

## Nonorthogonal three-waveguide coupler

In the following, we will look more closely into the nonorthogonal three-waveguide coupler, the simplest configuration in which the consequences of NOCMT projective non-Hermiticity can be observed while still allowing for an analytical solution of the corresponding differential equations. The simpler directional coupler is unsuitable because the Hamiltonians in OCMT and NOCMT are indistinguishable

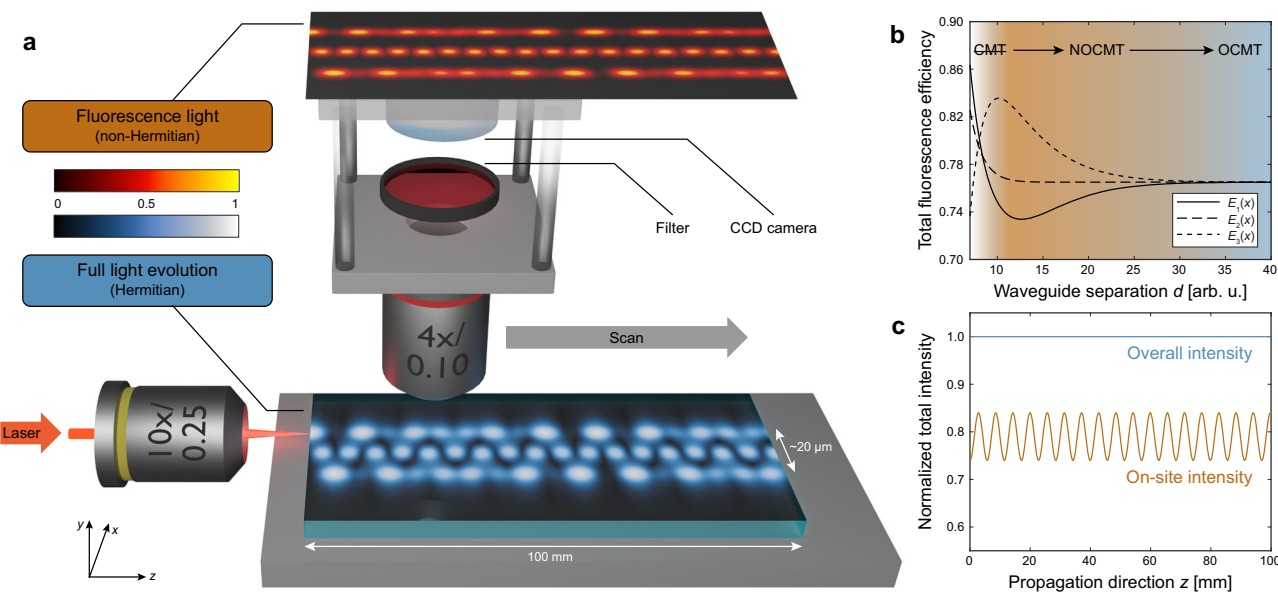

**Fig. 3 | Fluorescence microscopy as nonorthogonal projection. a** Light from a Helium-Neon laser (633 nm, cw) is injected through a microscope objective, whereupon the resulting wave packet evolves in the refractive index landscape of the coupled waveguide system. The simulated propagation pattern of the normalized field amplitude (split-step Fourier method) is shown in the blue-white colormap. Fluorescence from the waveguide cores is imaged perpendicularly onto a CCD, while scattered light is blocked by an appropriate laser line filter. A simulation of the resulting projective measurement is displayed at the top (red-orange colormap). **b** The total fluorescence efficiency with which light in a given supermode is imaged (cf. ratio between orange and dark orange area in Fig. 2j–l) depends on the degree of orthogonality, and therefore on the waveguide spacing. For well-

separated waveguides with orthogonal modes, the total fluorescence efficiency is identical for all three supermodes, i.e., the on-site fluorescence faithfully tracks the overall intensity with a constant scaling factor (OCMT = orthogonal coupled-mode theory). In contrast, under nonorthogonal mode conditions, the total fluorescence efficiency dynamically changes in line with the evolving interference pattern of the supermodes (NOCMT = nonorthogonal coupled-mode theory). **c** Quantitative comparison of the (conserved) overall intensity and the projectively measured on-site intensity. In the case of single-site excitation of one of the outer waveguides, the latter exhibits the characteristic sinusoidal oscillation of a non-Hermitian system with unbroken $\mathcal{PT}$-symmetry.

(see Supplementary Note 5). The resulting Hamiltonian reads as

$$H = \begin{pmatrix} 0 & \tilde{c} + \kappa\Delta & -\kappa\tilde{c} \\ \tilde{c} & \sigma - \kappa\tilde{c} & \tilde{c} \\ -\kappa\tilde{c} & \tilde{c} + \kappa\Delta & 0 \end{pmatrix}, \quad (2)$$

where $\tilde{c} = \frac{c - \delta_1 \kappa}{1 - 2\kappa^2}$ is the effective evanescent coupling coefficient, $\sigma = \beta_2 - \beta_1 + \frac{\delta_2 - \delta_1}{1 - 2\kappa^2}$ denotes the total detuning and $\Delta = \frac{\delta_1 - \delta_2}{1 - 2\kappa^2}$ is the contribution to the detuning stemming from self-couplings $\delta_k$[23]. The parameter $\kappa$ characterizes the nonorthogonality arising from the mode overlap between adjacent waveguides and vanishes in the OCMT limit. Being entirely real-valued, $H$ is non-Hermitian as long as the matrix is asymmetric, meaning that the key condition for non-Hermiticity is the nonorthogonality of the system ($\kappa > 0$). Beyond that, $H$ is invariant under complex conjugation, and $\mathscr{PT}$-symmetry is evidently fulfilled via $H = \mathscr{P} H \mathscr{P}$ with

$$\mathscr{P} = \begin{pmatrix} 0 & 0 & 1 \\ 0 & 1 & 0 \\ 1 & 0 & 0 \end{pmatrix}.$$

Figure 3a schematically illustrates the projective NOCMT measurement. Following the single-site excitation of one of the outer waveguides, the full evolution of the electric field $E(x, y, z)$, including the fraction residing in the interstitial regions, is calculated via the split-step Fourier method according to Eq. (3) (see "Methods") and displayed in a blue-shaded colormap. In accordance with the absence of gain or loss, for every point along the axis of propagation, its overall intensity is strictly constant (see Fig. 3c). In contrast, if only the light inside the waveguide cores is interrogated (red-shaded colormap), the total intensity distribution exhibits a pronounced sinusoidal oscillation (see Fig. 3c) in line with the inherent non-Hermiticity of the projective observation.

In general, a system of $N$ waveguides features up to $N$ distinct eigenvalues $\lambda_k$ that characterize the rate at which their associated modes accumulate phases during propagation. As the intensity is agnostic with respect to global phases, there may be as many as $\binom{N}{2} = N(N - 1)/2$ distinct frequency components arising from the beating between pairs of eigenmodes, while underlying symmetries of the lattice typically serve to constrain this number. The three eigenvalues of our three-waveguide coupler's Hamiltonian (2) give rise to the two frequencies (see Supplementary Note 9)

$$\omega_1 = \sqrt{2\tilde{c}^2 + \frac{\sigma^2}{4} + 2\tilde{c}\kappa\Delta}$$

and

$$\omega_2 = \frac{4\tilde{c}\kappa - \sigma}{2}$$

that, in turn, depend on the four physical parameters $\tilde{c}, \kappa, \sigma$ and $\Delta$. Up to a factor of 2, $\omega_1$ characterizes the overall range spanned by the first and third eigenvalues, and, due to its similarity to the behavior of the directional coupler, we will refer to it as "coupling frequency". In turn, $\omega_2$ describes the shift of the central eigenvalue from the symmetric position at the average of the outer ones. It is therefore termed "asymmetry frequency".

## Experimental results

For our experiments, we fabricated a set of twelve three-waveguide couplers with inter-waveguide spacings ranging from 7 to 18 μm, and conducted projective fluorescence measurements to determine the

systems' parameters via comparison to the analytical results. The upper part of Fig. 4a depicts the measured fluorescence pattern for the 10 μm system, while Fig. 4b shows the extracted intensities (solid black lines) as well as the best-fit analytical solution (red dashed lines). The non-Hermitian oscillation of the intensity sum periodically exceeds unity because of the initial condition that defines the total power at $z = 0$ as one (see Supplementary Note 6). The frequency $\omega_1$ can be directly identified from the sinusoidal oscillation of the central waveguide's intensity as well as the total intensity, whereas $\omega_2$ is responsible for the slower beating in the outer waveguides. With the values of these coefficients at hand, the Hamiltonian for the respective waveguide separation can be used to numerically compute the theoretical intensity dynamics for comparison (lower part of Fig. 4a). Along these lines, the dependence of the coupling coefficient $\tilde{c}$, the overlap $\kappa$ and the detuning $\sigma$ on the waveguide separation was tracked (Fig. 4c). Finally, the values of $\omega_{1,2}$ in the OCMT case can be estimated by setting the nonorthogonality parameter $\kappa = 0$. The deviation between the observed values and this Hermitian limit serves to quantify the degree of non-Hermiticity (Fig. 4d).

Note that for the structures probed in our experiments, all of these parameters decrease with growing waveguide separation (Fig. 4c). While the overlap $\kappa$ obviously has to vanish for large separations and the well-known exponential dependence of the coupling constant $\tilde{c}$ on the waveguide separation $d$[24] is routinely used for the precise design of specific coupling profiles[25], the reason behind the behavior of the detuning is at first glance unexpected. Intuitively, any detuning $\sigma$ should be independent of the waveguides' separation, as it is a measure for the refractive index profile of the cores. Note that, despite all three waveguides being inscribed with identical exposure parameters, the observed dynamics are indicative of substantial detunings $\sigma$. This observation illustrates the capability of fluorescence imaging to quantitatively characterize the influence of neighboring waveguides mediated through stress fields. In this vein, our measurements clearly track how these proximity effects become more pronounced in closely spaced settings, where the value of $\sigma$ exceeds $c$.

These intricacies notwithstanding, the system inevitably reaches the tight-binding limit (OCMT) as the overlap $\kappa$ vanishes for large separations. For the design of any experiment, the key question is, however, which conditions sufficiently preserve the convenient OCMT's validity while providing access to an effective propagation range over which meaningful wave dynamics can unfold. Clearly, the magnitude of the overlap itself is insufficient to make this determination, as illustrated by the relative deviations of the frequencies to their tight-binding limit

$$\frac{\Delta\omega_{\text{TB}_{1,2}}}{\omega_{1,2}} = \frac{|\omega_{1,2}(\exp) - \omega_{1,2}(\kappa = 0)|}{|\omega_{1,2}(\exp)|}$$

plotted logarithmically in Fig. 4d. While the actually observed coupling frequency $\omega_1$ remains within $10^{-1}$ (10%) of the tight-binding case over the entire range, the relative deviation of the asymmetry frequency $\omega_2$ exhibits a sharp exponential rise as for the separation is decreased. For separations below 11 μm, the nonorthogonality becomes the dominant influence as the relative deviation exceeds $10^0$ (100%). The deviation of $\omega_2$ approaches $10^{-1}$ (10%) for larger separations at 16 μm, indicating that the NOCMT regime can be readily accessed with an experimentally feasible choice of parameters, while the well-established OCMT model indeed yields reliable results at more conventional structural dimensions.

Naturally, the two descriptions can converge under different conditions, in particular, when only the total power guided in each core is probed (e.g., by observing the transverse intensity profile at the output facet). Likewise, the signatures of nonorthogonality may be hidden to an experimentalist on a fundamental level by the geometry of the structure itself, e.g., in the directional coupler, where the overlap

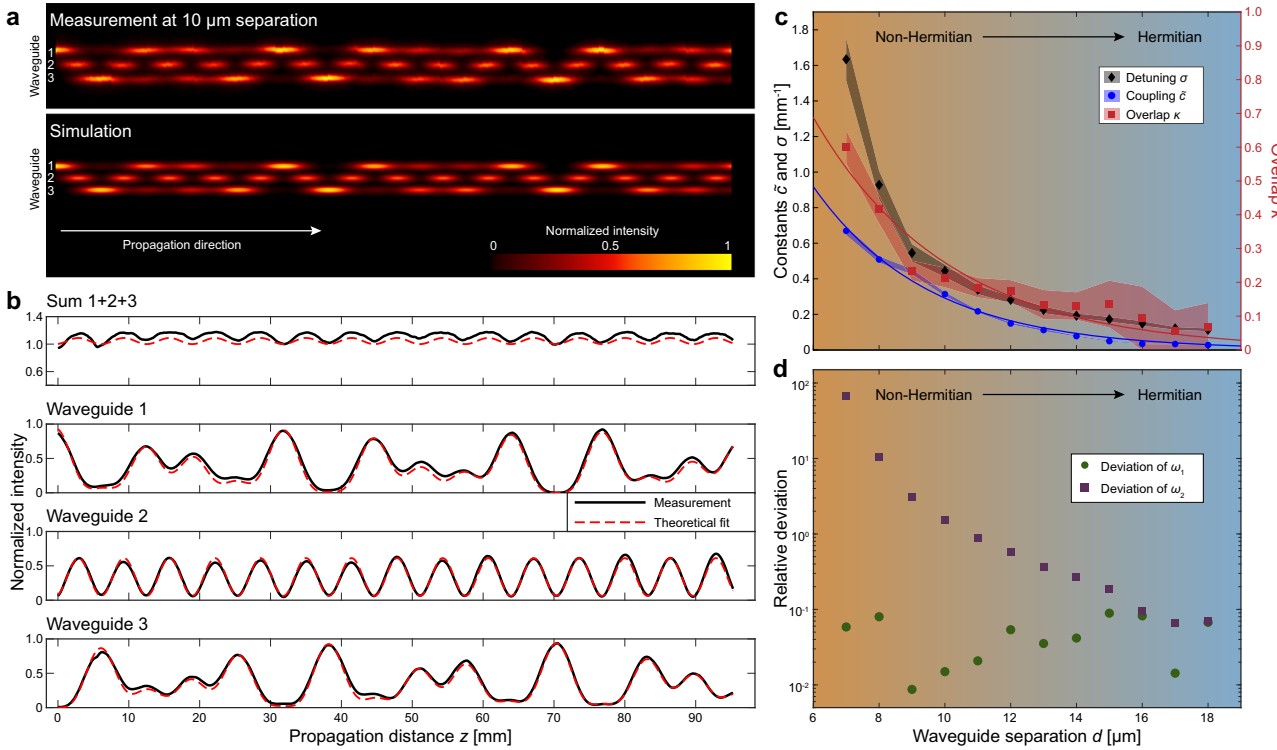

**Fig. 4 | Observing projective $\mathscr{PT}$-symmetry. a** Observed fluorescence pattern (top) and simulated on-site intensity distribution (bottom) in a three-waveguide coupler at a separation of 10 μm. **b** The sum (top) of the extracted on-site intensities of the three individual waveguides shows characteristic oscillatory dynamics in line with the unbroken $\mathscr{PT}$-symmetry established via the projective measurement. (Black solid lines: experimental data; red dashed lines: numerically calculated NOCMT dynamics according to $\tilde{c} = 0.3406\,\text{mm}^{-1}$, $\kappa = 0.1815$, $\sigma = 0.4226\,\text{mm}^{-1}$ and $\Delta = -0.3045\,\text{mm}^{-1}$). **c** Dependence of coupling $\tilde{c}$, overlap $\kappa$ and detuning $\sigma$ on the waveguide separation as determined from best-fit solutions for experiments at separations between 7 and 18 μm spacing. Note that while both coupling and detuning decay with increasing separation, the residual stress-induced detuning of

the central site becomes increasingly dominant. For the overlap $\kappa$ approaching zero at large separations, the orthogonal coupled-mode theory (tight-binding limit) can be utilized instead of the nonorthogonal one. **d** In order to construct a reliable indicator for the degree of nonorthogonality present in the system, we compare the relative deviations of the oscillation frequencies $\omega_1$ and $\omega_2$ from their tight-binding limit (i.e., setting $\kappa = 0$). The $\omega_1$ component can be directly identified from waveguide 2, whereas the asymmetry frequency $\omega_2$ manifests itself in the slower beating pattern in the outer waveguides (cf. (**b**)). Experiments conducted under conditions where the value of $\omega_2$ exceeds its uncertainty fall within the domain of NOCMT and therefore unequivocally establish non-Hermitian dynamics in the system at hand.

$\kappa$ does not appear as a measurable parameter but rather can be fully absorbed into effective coupling and detuning coefficients, respectively[26,27]. These caveats aside, the disparate dynamical length scales dictated by the inverse of the two frequencies $\omega_{1,2}$ allow for the full impact of the nonorthogonal modes to unfold whenever the evolution length $L$ approaches or exceeds the inverse of its spectral representatives, in our case, the asymmetry frequency, $1/\omega_2$.

## Discussion
In our work, we have shown how projective measurements in discrete systems governed by nonorthogonal coupled-mode theory can be leveraged to synthesize non-Hermitian and even $\mathscr{PT}$-symmetric systems without introducing any gain and loss, and demonstrated this approach experimentally via fluorescence imaging in laser-written waveguides. Our observations in three-waveguide couplers revealed the periodic variations of the total intensity characteristic of the unbroken phase of $\mathscr{PT}$-symmetric systems, as well as the emergence of a secondary beating period that is directly linked to the spectral asymmetry arising from partially overlapping nonorthogonal modes. Notably, projective $\mathscr{PT}$-symmetry is readily scalable to larger systems provided that the underlying refractive index profile is symmetric. The major advantages of our projective $\mathscr{PT}$-symmetry are the ability to design non-transposition-symmetric but nevertheless $\mathscr{PT}$-symmetric Hamiltonians and to explore non-Hermiticity also in contexts such as quantum optics, where implementations with actual gain/loss

arrangements are fundamentally precluded[13]. Beyond the obvious appeal of loss-free non-Hermiticity, projective non-Hermiticity is a general concept, allowing such designs to be applied to a wide variety of discrete platforms ranging from coupled cavities[28] and dielectric microcavities[29] to ultra-cold atoms in optical lattices[30,31] or acoustic and mechanical systems[32,33] as well as wave-mechanical arrangements with synthetic dimensions[34]. Given that fluorescence is inapplicable to these platforms, it is necessary to consider alternative techniques for applying a mask to the signal detection in order to perform the nonorthogonal projection, e.g., a localized detection in center of the traps for ultra-cold atoms, a localized measurement of the pressure in acoustics, or superconducting nanowire single-photon detection[35] (SNSPD) in the quantum optical context.

## Methods
### Coupled-mode theory
In general, the coherent light propagation in 2D waveguide arrays with small refractive index change is governed by the paraxial Helmholtz equation

$$\mathrm{i}\partial_z E(x,y,z) = -\left(\frac{1}{2n_0 k_0}\left[\partial_x^2 + \partial_y^2\right] + k_0 \Delta n(x,y)\right)E(x,y,z). \quad (3)$$

Any electric field $E(x,y,z)$ fulfilling this equation can be decomposed into a superposition of supermodes, which are the eigenmodes

$E_k(x,y)$ of the continuous Hamiltonian $\hat{H} = \frac{1}{2n_0 k_0}\left[\partial_x^2 + \partial_y^2\right] + k_0 \Delta n(x,y)$

$$\hat{H}E_k(x,y) = k_0(\lambda_k - n_0)E_k(x,y). \qquad (4)$$

The discrete nature of waveguide arrays allows for the expansion into tight-binding modes

$$E(x,y,z) = \sum_{k=1}^{N} a_k(z)w_k(x,y)e^{-i\beta_k z} \qquad (5)$$

with modal *amplitudes* $a_k(z)$ of the transverse *waveguide modes* $w_k(x,y)$ and propagation constants $\beta_k$. The waveguide modes $w_k(x,y)$ are defined via fulfilling their own Helmholtz Eq. (3) in the absence of the other waveguides and are thus localized parity-symmetrically around the $k$-th waveguide. As the shape of the waveguide modes does not change along the propagation direction, the area of a fixed-width slice around the maximum is proportional to its amplitude, such that $(a_k)_k$ can be read out experimentally via fluorescence microscopy. However, the waveguide modes do not necessarily form an orthonormal basis ($\int w_k^* w_j \mathrm{d}x\mathrm{d}y \neq \delta_{jk}$).

To address this issue, similarly to Eq. (5), alternative modal amplitudes $b_k(z)$ can be defined with respect to transverse *normal modes* $v_k(x,y)$ that form an orthonormal basis ($\int v_k^* v_j \mathrm{d}x\mathrm{d}y = \delta_{jk}$) under the trade-off of no possible corresponding measurement, such as fluorescence microscopy. For the three-waveguide coupler, the normal modes can be chosen in relation to the supermodes, e.g., via

$$\begin{pmatrix} v_1(x,y) \\ v_2(x,y) \\ v_3(x,y) \end{pmatrix} = \frac{1}{2}\begin{pmatrix} 1 & -\sqrt{2} & -1 \\ \sqrt{2} & 0 & \sqrt{2} \\ 1 & \sqrt{2} & -1 \end{pmatrix} \cdot \begin{pmatrix} E_1(x,y) \\ E_2(x,y) \\ E_3(x,y) \end{pmatrix} \qquad (6)$$

However, using the discretization (5) of Eq. (3), the light propagation is governed by the discrete Schrödinger Eq. (1), where the dynamics is now only determined by $a_k(z)$, while $w_k(x,y)$ defines the coupling constants $c_{jk} = k_0 \sum_{m\neq j}\int \Delta n_m w_k^* w_j \mathrm{d}x\mathrm{d}y$ (with the self-coupling $\delta_k = c_{kk}$, not to be confused with Kronecker delta) and the overlap $\kappa_{jk} = \int w_k^* w_j \mathrm{d}x\mathrm{d}y$. The Hamiltonian of our three-waveguide coupler (2) stems from the discrete Schrödinger Eq. (1) via $H = P^{-1}(C + PB)$ with the matrices

$$C = \begin{pmatrix} \delta_1 & c & 0 \\ c & \delta_2 & c \\ 0 & c & \delta_1 \end{pmatrix}, B = \begin{pmatrix} \beta_1 & 0 & 0 \\ 0 & \beta_2 & 0 \\ 0 & 0 & \beta_1 \end{pmatrix}, P = \begin{pmatrix} 1 & \kappa & 0 \\ \kappa & 1 & \kappa \\ 0 & \kappa & 1 \end{pmatrix}$$

under the assumption of only nearest-neighbor coupling and identical outer waveguides.

### Femtosecond laser direct writing

The photonic waveguide structures for our experiments were fabricated in fused silica (Corning 7980) by direct inscription[36] with ultra-short laser pulses from a frequency-doubled fiber amplifier system (Coherent Monaco 1035-80-60) at a wavelength of 517 nm, a repetition rate of 333 kHz and pulse energy of 3.24 μJ. Translating the focus of a 50x microscope objective (NA = 0.60) at a velocity of 100 mm/min through the volume of the 100 mm long glass chip results in a refractive index contrast of $\Delta n \approx 10^{-3}$, providing high-quality single-mode waveguides for the design wavelength of 633 nm with a slightly elliptical core with a cross-section of approximately 6 μm × 9 μm. In order to achieve symmetric arrangements, the central waveguide of the couplers was inscribed first, ensuring identical stress-induced detunings[37] for the two subsequently written outer channels.

### Fluorescence microscopy

The projective intensity measurements are enabled by the presence of non-bridging oxygen hole centers (NBOHCs) created inside the focal region during the inscription process[38]. These color centers serve to convert a small fraction of the guided light from a helium-neon laser (633 nm, continuous wave) into omnidirectional fluorescence at 650 nm, thereby providing insights into the propagation dynamics at any point along the chip[16]. In contrast to the conventional approach of directly projecting the end face of the chip onto a camera to observe the overall light distribution, imaging the color center fluorescence onto a CCD camera yields a quantitative measure for the fraction of the guided intensity contained in the core region of the waveguides. A laser line filter blocking the excitation wavelength of 633 nm serves to ensure high contrast images virtually without a scattered-light background.

## Data availability

The experimental data generated in this study have been deposited in the Rostock University Publication Server repository under accession code https://doi.org/10.18453/rosdok_id00004470[39]. Correspondence and requests for additional materials should be addressed to A.S.

## Code availability

The MATLAB® codes corresponding to the evaluation of the fluorescence measurements are available from the corresponding author (A.S.) upon request.

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

## Acknowledgements

We thank C. Otto for preparing the high-quality fused silica samples used for the inscription of all photonic structures employed in this work. A.S. acknowledges funding from the Deutsche Forschungsgemeinschaft (grants SZ 276/9-2, SZ 276/19–1, SZ 276/20–1, SZ 276/21–1, and SZ 276/27-1). A.S. also acknowledges funding from the Krupp von Bohlen and Halbach Foundation as well as from the FET Open Grant EPIQUS (grant no. 899368) within the framework of the European H2020 program for Excellent Science. A.S. and M.H. acknowledge funding from the Deutsche Forschungsgemeinschaft via SFB 1477 'Light–Matter Interactions at Interfaces' (project no. 441234705). A.S. and S.S. acknowledge funding from the Deutsche Forschungsgemeinschaft via GRK 2676/1-2023 'Imaging of Quantum Systems' (project no. 437567992).

## Author contributions

S.W. initiated the idea. L.J.M. and S.W. developed the experimental implementation. J.B. fabricated the samples, conducted the measurements, and together with M.H. evaluated the data. J.P. was consulted in theoretical discussions. A.S. and S.S. supervised the efforts of their respective groups. All authors discussed the results and co-wrote the manuscript.

## Funding

## Competing interests

The authors declare no competing interests.
