## [Transparent Peer review file · Nature Communications]

Unbroken PT-symmetry in the absence of gain or loss

Corresponding Author: Professor Alexander Szameit

Version 0:

Reviewer comments:

Reviewer #4

(Remarks to the Author)

The manuscript presents a novel approach for synthesizing non-Hermitian PT-symmetric optical devices that operate without the need for gain or loss. Traditional non-Hermitian or open systems typically rely on dissipation via intrinsic or particle losses and/or amplification through external gain. In contrast, the authors employ a non-orthogonal coupled mode theory to design non-Hermitian tight-binding models devoid of loss or gain, yet enabling the realization of distinctive features characteristic of unbroken PT-symmetric systems. Specifically, by carefully fabricating a system of three coupled waveguides—without any onsite loss or gain but incorporating nonreciprocal/asymmetric coupling—the authors have experimentally demonstrated PT-symmetric optical power oscillations and breathing dynamics due to non-orthogonal mode overlap. Fluorescence imaging technique was demonstrated as a non-orthogonal projection to probe the waveguide mode intensities. This discovery is intriguing and appealing to the non-Hermitian community. I am in favor of recommending the work for publication. Before that, I request the authors to please address or clarify the following concerns.

1. The use of "PT-symmetry" in the title is somewhat overstated and may be misleading given the limited scope of results presented in the manuscript. To clarify, the authors have effectively demonstrated gain-loss free non-Hermitian physics restricted exclusively to the regime of unbroken PT-symmetry. However, it is important to note that the domain of PT-symmetric non-Hermitian systems encompasses a broader range of phenomena, including PT-phase transitions, exceptional points, and broken PT-symmetry, none of which are addressed in this work. As one of the leading groups in this field, I am confident the authors recognize that significant advancements in non-Hermitian physics largely stem from investigations into non-Hermitian phase transitions—an aspect not yet clearly realized through non-orthogonal projection techniques as presented here. Therefore, I recommend that the authors revise the title to specifically reflect "Unbroken-PT-symmetry."
2. The non-Hermitian Hamiltonian matrix (2) exhibits a completely real spectrum. Is there a similarity transformation that accounts for this pseudo-Hermiticity?
3. The total power of the waveguide system periodically exceeds unity [Figure 4b]. Could the authors please clarify the origin of this power amplification in the absence of any gain mechanism?
4. Is the non-orthogonal projective method reversible? Specifically, is it possible to recover Hermitian physics from a deliberately designed non-Hermitian system? Kindly provide your insights on this matter.
5. The tight-binding lattice appears to require coupling between the first and last waveguides. Without an actual device image, it is unclear how this coupling was implemented.
6. It remains unclear how the proposed method can be generalized to systems with more than three waveguides. I recommend that the authors include additional matrix models involving a larger number of waveguides in the supplementary materials.
7. The tight-binding model bears resemblance to the Hatano-Nelson lattice. Would it be feasible, at least in principle, to demonstrate the skin effect using non-orthogonal fluorescence imaging techniques when lattice contains larger number of

waveguides ?

Version 1:

Reviewer comments:

Reviewer #4

(Remarks to the Author)

Authors have appropriately revised the manuscript as per my comments on earlier version.
Happy to recommend the revised manuscript for Nature Commun.

Response to Reviewers

Reviewer 4

Comment: *The manuscript presents a novel approach for synthesizing non-Hermitian PT-symmetric optical devices that operate without the need for gain or loss. Traditional non-Hermitian or open systems typically rely on dissipation via intrinsic or particle losses and/or amplification through external gain. In contrast, the authors employ a non-orthogonal coupled mode theory to design non-Hermitian tight-binding models devoid of loss or gain, yet enabling the realization of distinctive features characteristic of unbroken PT-symmetric systems. Specifically, by carefully fabricating a system of three coupled waveguides – without any onsite loss or gain but incorporating nonreciprocal/asymmetric coupling – the authors have experimentally demonstrated PT-symmetric optical power oscillations and breathing dynamics due to non-orthogonal mode overlap. Fluorescence imaging technique was demonstrated as a non-orthogonal projection to probe the waveguide mode intensities. This discovery is intriguing and appealing to the non-Hermitian community. I am in favor of recommending the work for publication. Before that, I request the authors to please address or clarify the following concerns.*

Response: We thank the reviewer for their endorsement and for providing constructive feedback. In the following, we have highlighted all revised sections in **yellow**.

Comment: *1. The use of "PT-symmetry" in the title is somewhat overstated and may be misleading given the limited scope of results presented in the manuscript. To clarify, the authors have effectively demonstrated gain-loss free non-Hermitian physics restricted exclusively to the regime of unbroken PT-symmetry. However, it is important to note that the domain of PT-symmetric non-Hermitian systems encompasses a broader range of phenomena, including PT-phase transitions, exceptional points, and broken PT-symmetry, none of which are addressed in this work. As one of the leading groups in this field, I am confident the authors recognize that significant advancements in non-Hermitian physics largely stem from investigations into non-Hermitian phase transitions—an aspect not yet clearly realized through non-orthogonal projection techniques as presented here. Therefore, I recommend that the authors revise the title to specifically reflect "Unbroken-PT-symmetry."*

Response: As suggested by the reviewer we changed the title of our revised manuscript to “**Unbroken** PT-symmetry in the absence of gain or loss”.

Comment: *2. The non-Hermitian Hamiltonian matrix (2) exhibits a completely real spectrum. Is there a similarity transformation that accounts for this pseudo-Hermiticity?*

Response: Our Hamiltonian is indeed pseudo-Hermitian, which can be seen when extending the proof of non-Hermiticity in the Supplementary information on page 2:

$$H^\dagger = (P^{-1}K)^\dagger = K^\dagger(P^{-1})^\dagger = KP^{-1} = KP^{-1}KK^{-1} = KHK^{-1}.$$

The aforementioned proof was incorporated in the revised Supplementary information:

“[...], which means the real-valued non-symmetric matrix H is necessarily non-Hermitian.

It is further pseudo-Hermitian [5] as it fulfills

$$H^\dagger = KP^{-1}KK^{-1} = KHK^{-1}.”$$

(Supplementary information, page 2, paragraph 1)

Comment: 3. *The total power of the waveguide system periodically exceeds unity [Figure 4b]. Could the authors please clarify the origin of this power amplification in the absence of any gain mechanism?*

Response: The power oscillation originates from the projective measurement in the nonorthogonal basis, or, in practical terms, in the fluorescence measurement that is selective to the waveguide cores. The total intensity of the light that is actually guided by the structure never exceeds unity, as there is no active gain in our system (see Fig. 3c, blue line). However, we selectively observe a specific part of our system when measuring the fluorescence emitted from the waveguide cores. The residual intensity remains invisible at each point of propagation (see Fig. 3c, orange line). In the case of the measured on-site intensity, the definition of unity determines whether the total power is periodically higher or lower than one. In our case, unity is defined through the initial condition when solving the differential equation ($A_1(0) + A_2(0) + A_3(0) = 1$, see Supplementary information page 8), which means the initial total power at $z = 0$ is set to one. We added the following sentence to our revised manuscript to clarify this issue:

“The non-Hermitian oscillation of the intensity sum periodically exceeds unity because of the initial condition that defines the total power at $z = 0$ as one (see Supplementary Chapter 6).”

(page 6, paragraph 2)

Comment: 4. *Is the non-orthogonal projective method reversible? Specifically, is it possible to recover Hermitian physics from a deliberately designed non-Hermitian system? Kindly provide your insights on this matter.*

Response: The nonorthogonal projective method is formally reversible when converting between the waveguide mode expansion (tight-binding model) and the normal mode expansion with the matrix Q . The transformation Q allows to recover a Hermitian Hamiltonian $\tilde{H} = (Q^{-1})^\dagger K Q^{-1}$ from the non-Hermitian Hamiltonian $H = P^{-1}K$. More information on that can be found in our manuscript on page 4 in paragraphs 2 and 3 and in Figure 2 as well as in chapter 2 of the Supplementary information. To highlight the reversibility of our method, we included the following statement in our revised Supplementary Information document:

“Because the matrix Q is invertible our nonorthogonal projective method is reversible.”

(Supplementary information, page 3, paragraph 1)

Comment: 5. *The tight-binding lattice appears to require coupling between the first and last waveguides. Without an actual device image, it is unclear how this coupling was implemented.*

Response: The coupling was implemented via the evanescent mode overlap between closely spaced laser-written waveguides (cf. equation (S3) in the Supplementary information on page 1). One can readily control the strength of the coupling constant via the separation between adjacent waveguides [24]. As coupling decays exponentially with the waveguide separation (cf. Fig. 4c, Fig. S6b and [24]), the dynamics in the system are quite accurately described by nearest-neighbor coupling, allowing us to neglect all higher-order couplings, including the one between the first and last waveguide. To emphasize this point we extended our Supplementary information carefully:

“Assuming symmetric coupling (for simplicity and because its exponential behavior is still obtained), equal outer waveguides (for \mathcal{PT} -symmetry) and neglecting ~~next-nearest-neighbor-coupling~~ higher-order couplings (because of the exponential dependence on the distance [6]), the matrices C , B and P look as follows [...]”

(Supplementary information, page 8, paragraph 1)

Comment: 6. It remains unclear how the proposed method can be generalized to systems with more than three waveguides. I recommend that the authors include additional matrix models involving a larger number of waveguides in the supplementary materials.

Response: We thank the author for this constructive suggestion. To take this into account we added the following chapter to the revised Supplementary Information document:

“7. NOCMT for more than three waveguides

Our method can easily be generalized to a system with more than three waveguides. The Hamiltonian can be calculated via $H = P^{-1}C + B$ with the propagation constants matrix B , the couplings matrix C , and the power overlap matrix P . With the reasonable assumptions of symmetric coupling and neglecting next-nearest-neighbor couplings together with the necessary condition of symmetric refractive index profile for \mathcal{PT} -symmetry the general matrices have the following form,

$$C = \begin{pmatrix} \delta_1 & c_1 & 0 & \cdots & 0 \\ c_1 & \delta_2 & c_2 & \ddots & \vdots \\ 0 & c_2 & \ddots & \ddots & 0 \\ \vdots & \ddots & \ddots & \delta_2 & c_1 \\ 0 & \cdots & 0 & c_1 & \delta_1 \end{pmatrix}, \quad B = \begin{pmatrix} \beta_1 & 0 & \cdots & \cdots & 0 \\ 0 & \beta_2 & \ddots & & \vdots \\ \vdots & \ddots & \ddots & \ddots & \vdots \\ \vdots & & \ddots & \beta_2 & 0 \\ 0 & \cdots & \cdots & 0 & \beta_1 \end{pmatrix}, \quad P = \begin{pmatrix} 1 & \kappa_1 & 0 & \cdots & 0 \\ \kappa_1 & 1 & \kappa_2 & \ddots & \vdots \\ 0 & \kappa_2 & \ddots & \ddots & 0 \\ \vdots & \ddots & \ddots & \ddots & \kappa_1 \\ 0 & \cdots & 0 & \kappa_1 & 1 \end{pmatrix}.$$

After calculating the Hamiltonian, one has to solve the discrete Schrödinger equation (S4) to get the intensity distribution of the single sites. The eigenvalue spectrum of the Hamiltonian yields insights to the different frequency components that determine the dynamics of the system. Of course, as the number of parameters increases with the number of sites, the analytical calculation gets more cumbersome and a numerical calculation may be more appropriate.”

(Supplementary information, page 9, paragraph 2)

Comment: 7. The tight-binding model bears resemblance to the Hatano-Nelson lattice. Would it be feasible, at least in principle, to demonstrate the skin effect using non-orthogonal fluorescence imaging techniques when lattice contains larger number of waveguides?

Response: The reviewer is correct, both the Hatano-Nelson model and the nonorthogonal coupled-mode theory accomplish asymmetric couplings. However, the stronger (and weaker) coupling in the Hatano-Nelson lattice is unidirectional, while it is alternating in the nonorthogonal coupled-mode theory (see figure below). Even if we increase the number of lattice sites in our system, we cannot achieve a similar coupling scheme to the Hatano-Nelson lattice as it is forbidden due to the condition of \mathcal{PT} -symmetry.